# Synthesis and Photolytic Assessment of Nitroindolinyl-Caged Calcium Ion Chelators [note 1]

**DOI:** 10.3390/molecules27092645

**Published:** 2022-04-20

**Authors:** George Papageorgiou, John E. T. Corrie

**Affiliations:** 1Chemical Biology STP, The Francis Crick Institute, 1 Midland Road, London NW1 1AT, UK; 2MRC National Institute for Medical Research, The Ridgeway, Mill Hill, London NW7 1AA, UK; jcorrie@nimr.mrc.ac.uk

**Keywords:** nitroindoline cages, calcium chelators, chemical synthesis, photolysis, photostability

## Abstract

Neuroactive amino acids derivatised at their carboxylate groups with a photolabile nitroindolinyl group are highly effective reagents for the sub-µs release of neuroactive amino acids in physiological solutions. However, the same does not apply in the case of calcium ion chelators. In this study, nitroindolinyl-caged BAPTA is found to be completely photostable, whereas nitroindolinyl-caged EDTA photolyses only when saturated with calcium ions.

## 1. Introduction

Localised fluctuations of free Ca^2+^ concentrations play critical roles in essential physiological processes [1,2,3,4,5], such as neurotransmission, skeletal and cardiac muscle contraction, hormone secretion, chemotaxis and blood clotting. In order to achieve sudden jumps of calcium ions in an experimental set up, a number of different photolabile molecules that bind Ca^2+^ have been developed over the last 35 years [6,7,8,9,10,11,12,13]. Flash irradiation of these compounds with near-UV light causes either fragmentation or other structural change, which results in a fast change in their calcium affinity, thereby resulting in the rapid release of Ca^2+^ ions in solution and triggering a biological response. This concept is illustrated by the first of these reagents, DM-nitrophen, which is a photolabile derivative of EDTA (Figure 1) [12]. However, despite the canonical scheme of the photolysis shown in Figure 1, it is notable that compounds incorporating an *N*-(2-nitrobenzyl)glycine moiety, as in DM-nitrophen, are susceptible to a side reaction of photo-decarboxylation. For DM-nitrophen, a detailed study showed this comprised approximately 10% of the total amount of the compound photolysed in the absence of Ca^2+^, and significantly more (16.5%) in the presence of saturating Ca^2+^ [14].

The advantages and drawbacks of the various reagents for the photorelease of Ca^2+^ developed over the years since the introduction of DM-nitrophen have been discussed [15] and the field can be considered to be well matured. In contrast, photochemical means to effect rapid decreases in Ca^2+^ concentration have been less well described, yet Ca^2+^ signalling is widespread and includes processes (reviewed by Berridge et al. [16] including muscle function, fertilisation, axis formation, cell differentiation, proliferation, transcriptional activation and apoptosis. Although much research has focuses on the initiation of processes due to a rise in the Ca^2+^ concentration, there must be a subsequent fall to reset the system. To our knowledge, the only reagents described to reduce Ca^2+^ concentrations experimentally are restricted to studies by the Tsien laboratory [17] and to a preliminary communication by Ferenczi et al. [18]. All of the reagents explored in these reports are based on photochemistry in which the calcium affinity of the chelator 1,2-bis(2-aminophenoxy)ethane-*N,N,N′,N′*-tetraacetic acid (BAPTA) is weakened by the incorporation of a photolabile modification. BAPTA itself has *K*_d_ ~0.11 µM at pH 7 and, in the strategies previously explored, blocking one of its chelating carboxylate groups weakens *K*_d_ by approximately three orders of magnitude [17,18]. A similar reduction is discussed below for the nitroindoline derivative of EDTA **24**. We aimed to extend this approach, using the 7-nitroindolinyl caging group previously developed by us for caging L-glutamate [19,20] and other amino acids (Figure 2) In this event, irradiation of the compounds synthesised showed surprisingly little photosensitivity and the results are described herein.

## 2. Results and Discussion

Our extended programme for the development of effective photocleavable cages of biologically active compounds included the investigation of the photolysis mechanism(s) of nitroindoline-caged compounds [21,22]. Having established that nitroindolinyl groups covalently linked to neuroactive amino acids can release biologically active compounds on a sub-microsecond time scale upon flash photolysis, we embarked on the synthesis of nitroindoline-caged calcium chelators that could potentially cause rapid calcium uptake upon photorelease. A summary of the results of this work has been disclosed in a review [23] and full details of the synthesis and photochemical evaluation are described here.

We envisaged that it would be optimal to construct the BAPTA framework and covalently attach the photolabile nitroindoline cage towards the end of the synthesis process. We initially planned the synthesis by employing our first developed nitroindoline cage bearing a CH_2_CO_2_Me group at the 5-position of the ring [19]. Thus, 2-nitrophenol was easily alkylated to 2-(2-nitrophenoxy)ethanol **1**, which was reduced to 2-(2-aminophenoxy)ethanol **2**. Double alkylation gave **3**, which was tosylated to **4** and converted to bromide **5**. Bromide displacement of **5** with 2-nitrophenol gave the desired compound **6** in a moderate yield (Figure 1).

However, two by-products **6a** and **6b** were also isolated from this reaction. We speculate that, in competition with the expected intermolecular bromide displacement by the 2-nitrophenoxide, **5** had also undergone an intramolecular cyclisation process to form a quaternary benzoxazonium species, with subsequent dealkylation by 2-nitrophenoxide to give by-products **6a** and **6b** (Figure 2).

Compound 6 was then reduced to aniline **7**. In straightforward steps, indoline **8** [19] was acylated to give the bromoacetyl indoline **9**, which, after nitration, gave the photolabile precursor **10**. Alkylation of amine **7** with **10** gave **11** but a number of attempts to achieve a final *N*-alkylation of **11** with *t*-butyl bromoacetate gave only a complex mixture of uncharacterised products (Figure 3).

To probe the failure of the required *N*-alkylation of the secondary amine of **11**, we prepared the less congested amine **12**. Treatment with a slight excess of *t*-butyl bromoacetate and either sodium hydride or DIPEA showed that *C*-alkylation on the 5-substituent of the indoline was the principal product **13**, whereas a large excess of the alkylating reagent and a higher reaction temperature gave the doubly alkylated product **14** (Figure 4).

To avoid the unwanted alkylation, we changed to a 5-methyl substituted indoline, and the synthesis then proceeded smoothly to give **17**, which, upon treatment with an excess of *t*-butyl bromoacetate, gave the protected caged BAPTA **18** in a reasonable yield. Surprisingly, we also isolated a substantial amount of the tetrabutyl ester of BAPTA **18a** (Figure 5). Evidently, hydrolysis by water, which formed from K_2_CO_3_ during alkylation, had caused some hydrolysis of the amide, and the released free carboxylate then underwent esterification to give **18a.** Deprotection with TFA successfully gave the target nitroindolinyl-caged BAPTA **19** (Figure 5).

Having prepared our target NI-caged BAPTA chelator, we next examined its near-UV photolysis properties. Surprisingly, sequential irradiations of a solution of **19** monitored via UV spectroscopy, as described in the Experimental Section, showed no change ascribable to the expected nitroindolinyl photocleavage, either with or without Ca^2+^ present (Figure 3). Control irradiation of NI-caged glutamate under identical conditions confirmed the expected photolysis, as previously reported (data not shown) [19].

It appears that the photostability of **19** is likely to arise from quenching of the excited state of the nitroindolinyl group by the electron-rich aryl ring(s).We have previously observed an apparently similar quenching in the electron-rich 1-acetyl-4-dimethylamino-7-nitroindoline [24] and efficient formation of a low-lying triplet state in *N,N*-dimethylamino-4-nitroaniline has been reported elsewhere [25].

After the failure to record any measurable photolysis of NI-caged BAPTA **19**, we changed our focus to the synthesis of MNI-caged EDTA. We chose to employ the more efficient 4-methoxy-7-nitroindolinyl (MNI) cage, previously developed in our laboratory [20,24]. Thus, nitration of *N*-bromoacetyl indoline **20** gave essentially only the 7-nitro isomer **21**, which reacted with secondary amine **22** to give the tri-*tert*-butyl ester **23**. Deprotection with TFA afforded the desired MNI-caged EDTA **24** (Figure 6).

We then exposed MNI-caged EDTA **24** to sequential near-UV irradiation in the absence (A) and presence (B) of Ca^2+^. Clean photolysis of **24** was only observed in solution (B), producing good isosbestic points and the characteristic appearance of the nitrosoindole photoproduct formation, as evident from the cumulative appearance of the new peak maximum at 413 nm (Figure 4). Irradiation in the absence of Ca^2+^ evidently caused photolytic changes, as shown by the progressive rise in absorption near 340 nm, but the product(s) were not investigated further. It is likely that, in the absence of Ca ions, irradiation of **24** can instigate electron transfer from the aliphatic tertiary amino groups to the excited state of the nitroindoline moiety and thus block the expected photolysis pathway. Such single-electron transfer is likely to trigger decarboxylation of **24**, as reported to occur for cation radicals of α-amino acids [14,26]. In solution (B), as the caged chelator has a *K*_d_ for calcium of ~10 µM (cf., the similar *K*_d_ value for the related pentadentate chelator *N*-methylethylenediaminetriacetic acid [27]), the large excess of Ca^2+^ would cause full saturation. Thus, the lone electron pairs on each of the tertiary amino groups would be unavailable for single-electron transfer and the normal 7-nitroindolinyl photolysis can operate, as shown by the appearance of the 413 nm band upon irradiation. Notably, the 5,7-dinitroindolinyl-caged BAPTA amide reported previously [17] exhibited the same photolytic behaviour as **24** and only photolysed in the presence of excess Ca^2+^.

## 3. Experimental Section

^1^H NMR spectra were determined on Varian Unityplus 500 or JEOL FX90Q spectrometers in CDCl_3_ solution with TMS as an internal reference, unless otherwise specified. Elemental analyses were carried out by MEDAC Ltd., Surrey, UK. Electrospray mass spectra were recorded at the School of Pharmacy, University of London. Merck 9385 silica gel was used for flash chromatography. Analytical HPLC was performed on a 250 mm × 4 mm Merck Lichrospher RP8 column or a 125 mm × 4 mm Whatman Partisphere SAX column. The flow rate was 1.5 mL min^−1^ with either column. Preparative HPLC was carried out on a 2 cm × 30 cm column (Waters C_18_ packing, Cat. No. 20594) at a 2 mL min^−1^ flow rate. Details of mobile phases are given at relevant points in the text. Triethylammonium bicarbonate (TEAB) solution was prepared by bubbling CO_2_ into an ice-cold aqueous solution of 1 M triethylamine until the pH stabilised (pH ~7.4). Preparative anion-exchange chromatography used a column of DEAE-cellulose (2 cm × 20 cm). Detection for all analytical and preparative chromatography was at 254 nm. Organic solvents were dried over anhydrous Na_2_SO_4_ and evaporated under reduced pressure. Hexanes (bp 40–60 °C) were redistilled before use. Photolysis experiments were performed in a Rayonet RPR-100 photochemical reactor fitted with 16 nm × 350 nm lamps.

### 3.1. 2-(2-Nitrophenoxy)ethanol ***1***

To a solution of 2-nitrophenol (6.95 g, 50 mmol) in acetone (200 mL) was added anhydrous K_2_CO_3_ (13.82 g, 100 mmol), NaI (0.5 g), and freshly distilled 2-bromoethanol (31.24 g, 250 mmol) and the mixture was heated under reflux for 8 h. After cooling to rt the solid was filtered, washed with fresh acetone and the filtrate concentrated in vacuo. The residue was dissolved in CH_2_Cl_2_, washed with 1 M aq. NaOH, brine, dried, and evaporated to give brown oil. Fractional distillation under reduced pressure afforded **1 [28]** (8.18 g, 89%) as a pale oil, bp 138 °C/0.5 mmHg.

### 3.2. 2-(2-Aminophenoxy)ethanol ***2***

A solution of 2-(2-nitrophenoxy)ethanol **1** (7.33 g, 40 mmol) in conc. HCl (100 mL) was cooled to 0–5 °C and treated portionwise over 1 h with freshly activated zinc powder (13.07 g, 200 mmol). The mixture was then stirred at rt for 2 h, before it was poured into ice-water (200 mL) and basified to pH 12 with 4 M aq. NaOH. The precipitated white solid was filtered, washed with water and the filtrate extracted with CH_2_Cl_2_. The combined organic phases were washed with brine, dried, and evaporated to give a light brown solid. Recrystallisation (EtOAc-hexanes) and decolourisation with charcoal afforded **2** (4.42 g, 72%) as white plates, mp 88–89 °C (lit. [28] mp 90.9 °C).

### 3.3. Di-tert-butyl 2,2-((2-(2-hydroxyethoxy)phenyl)azanediyl)diacetate ***3***

To a solution of 2-(2-aminophenoxy)ethanol **2** (3.06 g, 20 mmol) in MeCN (200 mL) was added anhydrous K_2_CO_3_ (5.53 g, 40 mmol) and *t*-butyl bromoacetate (9.75 g, 50 mmol) and the mixture was heated under reflux. The reaction progress was followed by TLC (EtOAc-hexanes (2:3)) and after 6 h more *t*-butyl bromoacetate (9.75 g) was added and reflux was continued for 20 h. After cooling to rt, the solid was filtered, washed with fresh MeCN, and the filtrate concentrated in vacuo. The residue was dissolved in EtOAc, washed with saturated aq. NaHCO_3_ and brine, dried, and evaporated to give colourless oil (8.88 g). Flash chromatography (EtOAc-hexanes (2:3)) afforded **3** (6.21 g, 81%) as a colourless viscous oil; ν_max_/cm^−1^ (Film) 3425, 2980, 2930, 1735, 1600, 1500, 1370, 1150, 745; ^1^H NMR (90 MHz) *δ* 6.86 (4H, br s, ArH), 4.06 (2H, t, *J* = 4.0 Hz, ArOC*H*_2_), 4.00 (4H, s, NCH_2_), 3.78 (2H, t, *J* = 4.0 Hz, C*H*_2_OH), 1.45 (18H, s, CMe_3_).

### 3.4. Di-tert-butyl 2,2-((2-(2-(tosyloxy)ethoxy)phenyl)azanediyl)diacetate ***4***

A solution of the alcohol **3** (4.70 g, 12.3 mmol) in dry pyridine (20 mL), cooled to 0–5 °C, was treated with *p*-tosyl chloride (4.69 g, 24.6 mmol) and stirred for 20 h, allowing the solution to reach an ambient temperature gradually. The mixture was poured into ice water and stirred at 0 °C for 2 h. The precipitated white solid was filtered, washed with cold water, and dried in a vacuum desiccator, affording **4** (5.15 g, 78%) as a white solid. A sample of the product was recrystallised from cold ether-hexanes to give white fine needles, mp 78–79 °C; ν_max_/cm^−1^ (Nujol) 1760, 1740, 1600, 1360, 1250, 1180, 1150, 940, 755; ^1^H NMR (500 MHz) *δ* 7.82 (2H, d, *J* = 8.5 Hz, H-2′ and H-6′), 7.33 (2H, d, *J* = 8.5 Hz, H-3′ and H-5′), 6.90 (1H, dt, *J* = 7.9, 1.2 Hz, H-5), 6.85 (2H, dt, *J* = 7.9, 1.2 Hz, H-4 and H-6), 6.74 (1H, dt, *J* = 7.3, 1.2 Hz, H-3), 4.34 (2H, t, *J* = 4.9 Hz, ArOC*H*_2_, 4.21 (2H, t, *J* = 4.9 Hz, C*H*_2_OSO2), 3.96 (4H, s, 2 × NCH_2_) 1.41 (18H, s, 2 × CMe_3_). Anal. Calcd for C_27_H_37_NO_8_S: C, 60.54; H, 6.96; N, 2.61; found: C, 60.36; H, 6.96; N, 2.57.

### 3.5. Di-tert-butyl 2,2-((2-(2-bromoethoxy)phenyl)azanediyl)diacetate ***5***

To a solution of the tosylate **4** (5.47 g, 10.2 mmol) in acetone (200 mL) was added lithium bromide (7.97 g, 91.8 mmol) and the mixture was stirred at rt for 18 h. The solvent was evaporated, and the residue was dissolved in water and washed with CH_2_Cl_2_. The combined organic phases were washed with saturated aq. NaHCO_3_ and brine, dried, and evaporated, affording **5** (4.53 g, 100%) as a light brown oil, which was used in the next step without further purification; ^1^H NMR (90 MHz) *δ* 6.87 (4H, br s, ArH), 4.30 (2H, t, *J* = 6.5 Hz, ArOC*H*_2_), 4.04 (4H, s, NCH_2_), 3.60 (2H, t, *J* = 6.5 Hz, CH_2_Br), 1.43 (18H, s, CMe_3_).

### 3.6. Di-tert-butyl 2,2-((2-(2-(2-nitrophenoxy)ethoxy)phenyl)azanediyl)diacetate ***6***

To a solution of the bromide **5** (1.02 g, 2.29 mmol) in acetone (25 mL) was added 2-nitrophenol (0.64 g, 4.58 mmol), NaI (20 mg), and anhydrous K_2_CO_3_ (1.27 g, 9.16 mmol) and the mixture was heated under reflux for 22 h. The solid was filtered, washed with acetone and the filtrate evaporated. The residue and the solid were dissolved in water and washed with ether. The combined organic phases were washed with 1 M aq. NaOH and brine, dried and evaporated. Flash chromatography (EtOAc-hexanes ((1:9)→(2:8)→(3:7))) gave three products. The first eluted material was *tert-butyl 2-(2,3-dihydro-4H-benzo[b][1,4]oxazin-4-yl)acetate*
**6a** (203 mg, 35%) as white crystals, mp 59–60 °C (hexanes); ν_max_/cm^−1^ (Nujol) 1740, 1610, 1585, 1370, 1350, 1310, 1260, 1240, 1215, 1150, 1110, 995, 745; ^1^H NMR (90 MHz) *δ* 6.88–6.40 (4H, m, ArH), 4.22 (2H, t, *J* = 5.0 Hz, H-2), 3.84 (2H, s, NCH_2_), 3.42 (2H, t, *J* = 5.0 Hz, H-3), 1.40 (9H, s, CMe_3_). Anal. Calcd for C_14_H_19_NO_3_: C, 67.45; H, 7.68; N, 5.62; found: C, 67.51; H, 7.76; N, 5.60.

The second eluted material was *tert*-butyl *2-(2-nitrophenoxy)acetate* **6b** (182 mg, 31%) as a viscous oil, which after treatment with TFA gave *2-(2-nitrophenoxy)acetic acid* (139 mg) as white crystals, mp 155–156 °C (EtOAc-hexanes), (lit. [29] 160–162 °C).

The third eluted material was **6** (527 mg, 46%) as white crystals, mp 101–102 °C (EtOAc-hexanes); ν_max_/cm^−1^ (Nujol) 1750, 1735, 1610, 1530, 1370, 1250, 1150, 745; ^1^H NMR (500 MHz) 7.84 (1H, dd, *J* = 8.1, 1.6 Hz, H-3′), 7.53 (1H, dt, *J* = 8.4, 1.8 Hz, H-5′), 7.17 (1H, dd, *J* = 8.6, 1.0 Hz, H-6′), 7.05 (1H, dt, *J* = 8.1, 1.1 Hz, H-4′), 6.96–6.87 (4H, m ArH), 4.47–4.42 (4H, m, OCH_2_), 4.02 (4H, s, NCH_2_), 1.40 (18H, s, CMe_3_). Anal. Calcd for C_26_H_34_N_2_O_8_: C, 62.14; H, 6.82; N, 5.57; found: C, 62.27; H, 6.86; N, 5.55.

### 3.7. Di-tert-Butyl 2,2-((2-(2-(2-aminophenoxy)ethoxy)phenyl)azanediyl)diacetate ***7***

A solution of **6** (2.01 g, 4 mmol) in a mixture of EtOH-EtOAc (2:1, 120 mL) was hydrogenated at rt and atmospheric pressure over 10% Pd-C (400 mg). Hydrogen uptake ceased within 30 min, the catalyst was filtered and the filtrate evaporated, affording **7** (1.55 g, 82%) as white crystals, mp 98–99 °C (EtOH); ν_max_/cm^−1^ (Nujol) 3470, 3390, 1750, 1735, 1600, 1505, 1375, 1225, 1150, 945, 745; ^1^H NMR (90 MHz) *δ* 6.88–6.60 (8H, m, ArH), 4.35 (4H, s, OCH_2_), 4.06 (4H, s, NCH_2_), 3.86 (2H, br s, NH_2_), 1.40 (18H, s, CMe_3_). Anal. Calcd for C_26_H_36_N_2_O_6_: C, 66.08; H, 5.68; N, 5.93; found: C, 66.12; H, 7.71; N, 5.92.

### 3.8. Methyl 2-(1-(2-bromoacetyl)indolin-5-yl)acetate ***9***

A solution of methyl 2-(indolin-5-yl)acetate **8** (1.25 g, 6.5 mmol; prepared as previously described [19]) in dry CH_2_Cl_2_ (40 mL) and cooled to −78 °C was treated under nitrogen with DIPEA (1.74 mL, 10 mmol) and a solution of bromoacetyl bromide (1.61 g, 8 mmol) in dry CH_2_Cl_2_ (20 mL) was added dropwise over 1 h. The solution was stirred at −78 °C for 1 h, then allowed to warm to rt, diluted with CH_2_Cl_2_ and washed successively with 1 M aq. HCl, saturated aq. NaHCO_3_ and brine, dried and evaporated to give a brown oil, which crystallised after trituration with ether affording **9** (1.32 g, 65%) as white crystals, mp 86–87 °C (EtOAc-hexanes); ν_max_/cm^−1^ (Nujol) 1745, 1730, 1670, 1380, 1205, 1170; ^1^H NMR (90 MHz) *δ* 8.12 (1H, d, *J* = 8.1 Hz, H-7), 7.28–6.96 (2H, m, H-4 and H-6), 4.16 (2H, t, *J* = 8.1 Hz, H-2), 3.92 (2H, s, CH_2_Br), 3.66 (3H, s, CO_2_Me), 3.56 (2H, s, C*H*_2_CO_2_Me), 3.14 (2H, t, *J* = 8.1 Hz, H-3). Anal. Calcd for C_13_H_14_BrNO_3_: C, 50.02; H, 4.52; N, 4.49; found: C, 50.21; H, 4.70; N, 4.44.

### 3.9. Methyl 2-(1-(2-bromoacetyl)-7-nitroindolin-5-yl)acetate ***10***

To a stirred solution of NaNO_3_ (221 mg, 2.6 mmol) in TFA (12 mL) was added **9** (749 mg, 2.4 mmol) and the mixture was stirred at rt for 4 h. The red/brown solution was poured into ice-cold water and extracted with EtOAc. The combined organic phases were washed with saturated aq. NaHCO_3_, brine, dried and evaporated to give a red viscous oil, which, after trituration with Et_2_O, afforded **10** (679 mg, 79%) as yellow crystals, mp 102–103 °C (EtOAc-hexanes); ν_max_/cm^−1^ (Nujol) 1730, 1670, 1530, 1395, 1375, 1270, 1015; ^1^H NMR (90 MHz) *δ* 7.52 (1H, br s, H-6), 7.36 (1 H, br s, H-4), 4.28 (2H, t, *J* = 8.1 Hz, H-2), 3.92 (2H, s, CH_2_Br), 3.68 (3H, s, CO_2_Me), 3.62 (2H, s, C*H*_2_CO_2_Me), 3.22 (2H, t, *J* = 8.1 Hz, H-3). Anal. Calcd for C_13_H_13_BrN_2_O_5_: C, 43.72; H, 3.67; N, 7.84; found: C, 43.60; H, 3.62; N, 7.84.

### 3.10. Di-tert-butyl 2,2-((2-(2-(2-((2-(5-(2-methoxy-2-oxoethyl)-7-nitroindolin-1-yl)-2-oxoethyl)amino)phenoxy)ethoxy)phenyl)azanediyl)diacetate ***11***

A mixture of the aniline **7** (354 mg, 0.75 mmol), methyl 1-bromoacetyl-7-nitroindoline-5-acetate **10** (268 mg, 0.75 mmol) and DIPEA (97 mg, 0.75 mmol) in dry MeCN (15 mL) was heated under reflux in a nitrogen atmosphere for 4 h. After cooling to rt, the solvent was evaporated and the residue dissolved in EtOAc was washed with 100 mM Na phosphate, pH 5.0. The organic phase was dried and evaporated to give a brown oil. Flash chromatography (EtOAc-hexanes (1:1)) afforded **11** (446 mg, 79%) as yellow crystals, mp 129–130 °C (EtOAc-hexanes); ν_max_/cm^−1^ (Nujol) 3430, 1740, 1680, 1605, 1530, 1380, 1145, 745; ^1^H NMR (500 MHz) *δ* 7.55 (1H, s, H-6), 7.34 (1H, s, H-4), 6.94-6-86 (4H, m, ArH), 6.83-6-81 (2H, m, ArH), 6.69 (1H, dt, *J* = 7.7, 1.1 Hz, ArH), 6.62 (1H, dd, *J* = 7.7, 1.1 Hz, ArH), 5.21 (1H, t, *J* = 5.1 Hz, NH), 4.39–4.36 (4H, m, ArOCH_2_), 4.26 (2H, t, *J* = 8.1 Hz, H-2), 4.11 (2H, d, *J* = 5.1 Hz, NCH_2_CON), 4.06 (4H, br s, NCH_2_), 3.70 (3H, s, CO_2_Me), 3.62 (2H, s, C*H*_2_CO_2_Me), 3.11 (2H, t, *J* = 8.1 Hz, H-3), 1.41 (18H, s, CMe_3_). Anal. Calcd for C_39_H_48_N_4_O_11_: C, 62.55; H, 6.46; N, 7.48; found: C, 62.56; H, 6.57; N, 7.41.

### 3.11. Methyl 2-(1-((2-methoxyphenyl)glycyl)-7-nitroindolin-5-yl)acetate ***12***

A mixture of methyl 1-bromoacetyl-7-nitroindoline-5-acetate **10** (832 mg, 2.3 mmol), *o*-anisidine (283 mg, 2.3 mmol), and DIPEA (297 mg, 2.3 mmol) in dry MeCN (50 mL) was heated under reflux in a nitrogen atmosphere for 3 h. After cooling to rt, the solvent was evaporated and the residue was dissolved in CH_2_Cl_2_ and washed with aq. 100 mM Na phosphate, pH 5.0. The organic phase was dried and evaporated to give a brown oil. Flash chromatography (EtOAc-hexanes (3:2)) afforded **12** (735 mg, 80%) as yellow needles; mp 166–167 °C (EtOAc-hexanes); ν_max_/cm^−1^ (Nujol) 3420, 1750, 1735, 1675, 1600, 1530, 1155, 745; ^1^H NMR (500 MHz) *δ* 7.57 (1H, br s, H-6), 7.39 (1H, br s, H-4), 6.87 (1H, dt, *J* = 7.3, 1.5 Hz, H-5′), 6.78 (1H, dd, *J* = 7.8, 1.5 Hz, H-3′), 6.71 (1H, dt, *J* = 7.3, 1.5 Hz, H-4′), 6.53 (1H, dd, *J* = 7.8, 1.5 Hz, H-6′), 5.11 (H, t, *J* = 4.9 Hz, NH), 4.29 (2H, t, *J* = 8.1 Hz, H-2), 4.07 (2H, d, *J* = 4.9 Hz, CH_2_N), 3.85 (3H, s, OMe), 3.71 (3H, s, CO_2_Me), 3.64 (2H, s, C*H*_2_CO_2_Me), 3.25 (2H, t, *J* = 8.1 Hz, H-3). Anal. Calcd for C_20_H_21_N_3_O_6_: C, 60.14; H, 5.30; N, 10.02; found: C, 60.11; H, 5.28; N, 10.40.

### 3.12. 4-(tert-Butyl) 1-methyl 2-(1-((2-methoxyphenyl)glycyl)-7-nitroindolin-5-yl)succinate ***13***

To a solution of **12** (80 mg, 0.2 mmol) in dry DMF (5 mL) was added sodium hydride (60% dispersion in mineral oil; 8 mg, 0.2 mmol) and *t*-butyl bromoacetate (43 mg, 0.22 mmol) and the mixture was refluxed under nitrogen. The progress of the reaction was followed by TLC (EtOAc-hexanes (3:2)). After 3 h more *t*-butyl bromoacetate was added (43 mg) and the heating continued for further 1 h. The solution was diluted with water (50 mL) and washed with EtOAc. The combined organic phases were washed with saturated aq. NaHCO_3_, brine, dried, and evaporated to give brown oil. Flash chromatography (EtOAc-hexanes (1:1)) gave **13** (51 mg, 50%) as a pale oil; ^1^H NMR (500 MHz) *δ* 7.58 (1H, d, *J* = 1.4 Hz, H-6), 7.39 (1H, d, *J* = 1.4 Hz, H-4), 6.86 (1H, dt, *J* = 7.3, 1.3 Hz, H-5′), 6.77 (1H, dd, *J* = 7.3, 1.3 Hz, H-3′), 6.71 (1H, dt, *J* = 7.3, 1.5 Hz, H-4′), 6.53 (1H, dd, *J* = 7.8, 1.5 Hz, H-6), 5.10 (1H, br s, NH), 4.28 (2H, t, *J* = 8.1 Hz, H-2), 4.07 (2H, d, *J* = 2.9 Hz, COC*H*_2_NH), 4.05 (1H, dd, *J* = 9.5, 6.0 Hz, CH_2_C*H*CO_2_Me), 3.81 (3H, s, OMe), 3.68 (3H, s, CO_2_Me), 3.23 (2H, t, *J* = 8.1 Hz, H-3), 3.09 (1H, dd, *J* = 16.7, 9.5 Hz, half of C*H*_2_CHCO_2_Me), 2.61 (1H, dd, *J* = 16.7, 6.0 Hz, half of C*H*_2_CHCO_2_Me), 1.41 (9H, s, CMe_3_).

### 3.13. 4-(tert-Butyl) 1-methyl 2-(1-(N-(2-(tert-butoxy)-2-oxoethyl)-N-(2-methoxyphenyl)glycyl)-7-nitroindolin-5-yl)succinate ***14***

To a solution of **12** (100 mg, 0.25 mmol) in dry MeCN (10 mL) was added anhydrous K_2_CO_3_ (138 mg, 1 mmol) and *t*-butyl bromoacetate (488 mg, 2.5 mmol) and the mixture was heated under reflux. After 2 h, more *t*-butyl bromoacetate was added (488 mg, 2.5 mmol) and the heating continued for a further 16 h. The solid was filtered off, washed with MeCN and the filtrate evaporated. The residue was dissolved in EtOAc and washed with saturated aq. NaHCO_3_, brine, dried, and evaporated to give a brown oil. Flash chromatography (EtOAc-hexanes (2:3)) gave **14** (70 mg, 44%) as a pale oil; ^1^H NMR (500 MHz) *δ* 7.57 (1H, br s, H-6), 7.36 (1H, br s, H-4), 6.96 (2H, dt, *J* = 7.3, 1.3 Hz, H-5′ and H-3′), 6.90 (1H, dt, *J* = 7.3, 1.3 Hz, H-4′), 6.84 (1H, dd, *J* = 7.3, 1.5 Hz, H-6′), 4.30 (2H, s, NCH_2_CO_2_), 4.28 (2H, t, *J* = 8.1 Hz, H-2), 4.02 (1H, dd, *J* = 9.5, 6.0 Hz, CH_2_C*H*CO_2_Me), 3.98 (2H, s, NCOCH_2_N), 3.82 (3H, s, OMe), 3.68 (3H, s, CO_2_Me), 3.13 (2H, t, *J* = 8.1 Hz, H-3), 3.09 (1H, dd, *J* = 16.7, 9.5 Hz, half of C*H*_2_CHCO_2_Me), 2.59 (1H, dd, *J* = 16.7, 6.0 Hz, half of C*H*_2_CHCO_2_Me), 1.42 (9H, s, CMe_3_), 1.41 (9H, s, CMe_3_).

### 3.14. 2-Bromo-1-(5-methylindolin-1-yl)ethan-1-one ***15***

To an ice-cold solution of 5-methylindole (1.31 g, 10 mmol) in acetic acid (50 mL) NaCNBH_3_ (2.48 g, 30 mmol) was added portionwise over 10 min (exothermic reaction) and the mixture was then stirred at rt for 0.5 h. Water (2–3 mL) was added and the solvent removed in vacuo. The residue was dissolved in EtOAc and washed with saturated aq. NaHCO_3_, brine, dried, and evaporated to give *5-methylindoline* (1.33 g, 100%) as a light brown oil, which was immediately used in the next step; ^1^H NMR (90 MHz) *δ* 7.08–6.64 (2H, m, H-4 and H-6), 6.48 (1H, d, *J* = 8.1 Hz, H7), 3.48 (2H, t, *J* = 7.2 Hz, H-2), 3.20 (1H, s, NH exchangeable with D_2_O), 2.95 (2H, t, *J* = 7.2 Hz, H-3), 2.22 (3H, s, Me). The crude *5-methylindoline* was dissolved in dry CH_2_Cl_2_ (50 mL), cooled to −78 °C, and treated under nitrogen with DIPEA (2.1 mL, 12 mmol), and a solution of bromoacetyl bromide (2.22 g, 11 mmol) in dry CH_2_Cl_2_ (50 mL) was added dropwise over 1 h. The solution was stirred at −78 °C for 1 h, then allowed to warm to rt, diluted with CH_2_Cl_2_ and washed successively with 1 M aq. HCl, saturated aq. NaHCO_3_ and brine, dried, and evaporated to give **15** (2.14 g, 84%) as pale yellow crystals, mp 116–118 °C (EtOAc-hexanes); ν_max_/cm^−1^ (Nujol) 1650, 1595, 1490, 820, 645; ^1^H NMR (90 MHz) *δ* 8.00 (1H, d, *J* = 8.1 Hz, H-7), 7.08–6.76 (2H, m, H-4 and H-6), 4.15 (2H, t, *J* = 7.2 Hz, H-2), 3.91 (2H, s, CH_2_Br), 3.17 (2H, t, *J* = 7.2 Hz, H-3), 2.30 (3H, s, Me). Anal. Calcd for C_11_H_12_BrNO: C, 51.99; H, 4.76; N, 5.52; found: C, 51.77; H, 4.81; N, 5.34.

### 3.15. 2-Bromo-1-(5-methyl-7-nitroindolin-1-yl)ethan-1-one ***16***

A solution of **15** (2.03 g, 8 mmol) in TFA (30 mL) was cooled to 0–5 °C and treated with NaNO_3_ (0.75 g, 8.8 mmol), and the mixture was stirred at that temperature for 2 h. The dark red/brown solution was poured into ice-cold water and extracted with EtOAc. The combined organic phases were washed with saturated aq. NaHCO_3_ and brine, dried, and evaporated to give **16** (0.83 g, 35%) as yellow crystals, mp 165–166 °C (EtOAc-hexanes-charcoal); ν_max_/cm^−1^ (Nujol) 1670, 1530, 1340, 1225, 865, 780, 645; ^1^H NMR (90 MHz) *δ* 7.45 (1H, br s, H-6), 7.26 (1H, s, H-4), 4.31 (2H, t, *J* = 7.2 Hz, H-2), 3.95 (2H, s, CH_2_Br), 3.20 (2H, t, *J* = 7.2 Hz, H-3), 2.38 (3H, s, Me). Anal. Calcd for C_11_H_11_BrN_2_O_3_: C, 44.17; H, 3.71; N, 9.37; found: C, 44.46; H, 3.74; N, 9.18.

### 3.16. Di-tert-butyl 2,2-((2-(2-(2-((2-(5-methyl-7-nitroindolin-1-yl)-2-oxoethyl)amino)phenoxy)ethoxy)phenyl)azanediyl)diacetate ***17***

A mixture of aniline **7** (473 mg, 1 mmol), 1-bromoacetyl-5-methyl-7-nitroindoline **16** (299 mg, 1 mmol) and DIPEA (129 mg, 1 mmol) in dry MeCN (20 mL) was heated under reflux in a nitrogen atmosphere for 4 h. After cooling to rt, the solvent was evaporated and the residue was dissolved in EtOAc and washed with aq. 100 mM Na phosphate, pH 5.0. The organic phase was dried and evaporated to give brown oil. Flash chromatography (EtOAc-hexanes (1:1)) afforded **17** (478 mg, 69%) as yellow crystals, mp 139–141 °C (EtOAc-hexanes); ν_max_/cm^−1^ (Nujol) 3420, 1740, 1670, 1600, 1530, 1370, 1245, 1140, 745; ^1^H NMR (500 MHz) *δ* 7.45 (1H, d, *J* = 0.5 Hz, H-6), 7.21 (1H, d, *J* = 0.5 Hz, H-4), 6.95-6-86 (4H, m, ArH), 6.83-6-81 (2H, m, ArH), 6.68 (1H, dt, *J* = 7.7, 1.5 Hz, ArH), 6.62 (1H, dd, *J* = 7.7, 1.5 Hz, ArH), 5.21 (1H, br s, NH), 4.40–4.35 (4H, m, ArOCH_2_), 4.24 (2H, t, *J* = 8.1 Hz, H-2), 4.10 (2H, s, NCH_2_CON), 4.06 (4H, br s, NCH_2_), 3.08 (2H, t, *J* = 8.1 Hz, H-3), 2.35 (3H, s, Me), 1.41 (18H, s, CMe_3_). Anal. Calcd for C_37_H_46_N_4_O_9_: C, 64.33; H, 6.71; N, 8.11; found: C, 63.93; H, 6.71; N, 7.96.

### 3.17. Di-tert-butyl 2,2-((2-(2-(2-((2-(tert-butoxy)-2-oxoethyl)(2-(5-methyl-7-nitroindolin-1-yl)-2-oxoethyl)amino)phenoxy)ethoxy)phenyl)azanediyl)diacetate ***18***

To a solution of **17** (345 mg, 0.5 mmol) in dry MeCN (15 mL) was added anhydrous K_2_CO_3_ (276 mg, 2 mmol) and *t*-butyl bromoacetate (975 mg, 5 mmol) and the mixture was heated under reflux. After 2 h more *t*-butyl bromoacetate was added (975 mg) and the heating continued for a further 16 h. The solid was filtered off, washed with MeCN and the filtrate evaporated. The residue was dissolved in EtOAc and washed with saturated aq. NaHCO_3_ and brine, dried, and evaporated to give a brown oil. Flash chromatography (EtOAc-hexanes (2:3)) gave two products. The first eluted material, a viscous oil which crystallised after trituration with Et_2_O-hexanes, identified as tetra-*tert*-butyl 1,2-bis(2-aminophenoxy)ethane-*N,N,N′,N′*-tetraacetate **18a** (132 mg, 38%), white crystals, mp 113–114 °C (hexanes); ν_max_/cm^−1^ (Nujol) 1755, 1740, 1720, 1505, 1250, 1035, 755; ^1^H NMR (90 MHz) *δ* 6.68 (8H, br s, ArH), 4.35 (4H, br s, ArOCH_2_), 4.06 (8H, br s, NCH_2_), 1.41 (36H, s, CMe_3_). Anal. Calcd for C_38_H_56_N_2_O_10_: C, 65.12; H, 8.05; N, 4.00; found: C, 65.27; H, 8.14; N, 3.98.

The second eluted material was **18** (163 mg, 40%) which was isolated as a brown film and used in the next step without further purification; ^1^H NMR (90 MHz) 7.39 (1H, br s, H-6), 7.02 (1H, br s, H-4), 6.96–6.56 (8H, m, ArH), 4.33 (4H, br s, ArOCH_2_), 4.12 (2H, t, *J* = 8.0 Hz, H-2), 4.06 (2H, s, NCH_2_CO), 4.02 (6H, s, NCH_2_), 2.86 (2H, t, *J* = 8.0 Hz, H-3), 2.31 (3H, s, Me), 1.40 (27H, s, CMe_3_).

### 3.18. 2,2-((2-(2-(2-((Carboxymethyl)(2-(5-methyl-7-nitroindolin-1-yl)-2-oxoethyl)amino)phenoxy)ethoxy)phenyl)azanediyl)diacetic acid (NI-Caged BAPTA) ***19***

A solution of the crude brown film **18** (163 mg, 0.202 mmol) in TFA (10 mL) was stirred at rt for 4 h. The red-brown solution was concentrated in vacuo and the residue dissolved in water (80 mL). The pH was raised from 1.92 to 7.2, via the careful addition of 1 M aq. NaOH, and extracted with ether (3 × 80 mL). The aqueous solution (89 mL) was first concentrated in vacuo and then filtered through a 0.2 mm cellulose membrane and analysed via reverse-phase HPLC (mobile phase 25 mM Na phosphate, pH 6.0 + 30% MeCN at 1.5 mL/min). A major peak eluted at *t*_R_ 5.4 min and a minor peak at *t*_R_ 6.4 min. The solution was made up to 25 mM Na phosphate, pH 6.0, and loaded onto a preparative HPLC column. The column was eluted first with 25 mM Na phosphate, pH 6.0 for 1 h (all flow rates 1.5 mL/min) and then with 25 mM Na phosphate, pH 6.0 + 30% MeCN. Fractions containing pure product were analysed, combined, and quantified via UV spectroscopy: λ_max_ (25 mM Na phosphate, pH 7.0/nm 345 (ε/M^−1^cm^−1^ 2700)) to give **19** (*NI-**caged BAPTA)* (48 μmol). The contaminated fractions were re-analysed via standard anion exchange HPLC (mobile phase 50 mM ammonium phosphate, pH 6.0 + 10% MeCN at 1.5 mL/min), showing a minor peak at *t*_R_ 2.8 min and a major peak at *t*_R_ 6.0 min. The solution was then concentrated, and the residue dissolved in water (87 mL) and the pH was adjusted to 7.42. The aqueous solution was then purified via anion-exchange chromatography using a linear gradient formed from 10 to 500 mM TEAB (each 250 mL). Fractions containing the product, which were eluted at ~240 mM TEAB were analysed as above. Pure fractions were combined, concentrated, and re-evaporated from MeOH. The residue was dissolved in water (1 mL) and quantified via UV (11.2 mM, 11 μmol). The total yield of isolated pure product **19** was 59 μmol (29%). A portion of the product was exchanged to sodium salt with Dowex-50. The sodium salt had ^1^H NMR (500 MHz D_2_O; acetone ref.) 7.45 (1H, br s, H-6), 7.23 (1H, br s, H-4), 7.12–7.07 (1H, m, ArH), 7.04–6.94 (2H, m, ArH), 6.94–6.86 (1H, m, ArH), 6.86–6.75 (2H, m, ArH), 6.52–6.58 (1H, m, ArH), 6.45–6.52 (1H, m, ArH), 4.37 (4H, br s, ArOC*H*_2_), 4.20–4.16 (2H, m, NC*H*_2_CO_2_), 4.00 (2H, t, *J* = 7.9 Hz, H-2), 3.82 (2H, s, NCH_2_CON), 3.80 (4H, br s, NCH_2_), 2.86 (2H, t, *J* = 7.9 Hz, H-3), 2.36 (3H, s, Me).); LRMS (FAB): Calcd for (C_31_H_29_N_4_O_11_ + 2H)^−^: 635; found: 635.

### 3.19. 2-Bromo-1-(4-methoxy-7-nitroindolin-1-yl)ethan-1-one ***21***

To a well-stirred suspension of claycop (1.28 g) in a mixture of CCl_4_ (8 mL) and Ac_2_O (4 mL) was added 1-bromoacetyl-4-methoxyindoline **20** (0.54 g, 2 mmol; prepared as previously described [30]), and the mixture was stirred at room temperature for 4 h. The mixture was diluted with EtOAc (60 mL) and washed with saturated aq. NaHCO_3_ and brine, dried, and evaporated to yellow crystals. Recrystallisation (EtOAc) gave **21** (0.45 g, 72%) as fine yellow crystals, mp 172–173 °C (EtOAc-hexanes); ^1^H NMR (90 MHz, *d*_6_-DMSO) *δ* 7.70 (1H, t, *J* = 9.0 Hz, H-6), 6.68 (1H, t, *J* = 9.0 Hz, H-5), 4.26 (2H, t, *J* = 8.1 Hz, H-2), 4.32 (2H, s, CH_2_Br), 3.92 (3H, s, OMe), 3.06 (2H, t, *J* = 8.1 Hz, H-3). Anal. Calcd for C_11_H_11_BrN_2_O_4_: C, 41.93; H, 3.52; N, 8.89; found: C, 42.16; H, 3.61; N, 8.62.

### 3.20. Di-tert-butyl 2,2-((2-((2-(tert-butoxy)-2-oxoethyl)(2-(4-methoxy-7-nitroindolin-1-yl)-2-oxoethyl)amino)ethyl)azanediyl)diacetate ***23***

A mixture of **21** (156 mg, 0.49 mmol), *N,N,N′*-tris(*tert*-butyloxycarbonylmethyl)ethane-1,2-diamine **22** (219 mg, 0.54 mmol, prepared as previously described [31]) and diisopropylethylamine (84 mg, 0.65 mmol) in dry MeCN (15 mL) was stirred at room temperature under a nitrogen atmosphere for 24 h. The precipitated white solid was filtered off and the filtrate was evaporated. The residue was dissolved in EtOAc (50 mL), washed with saturated aq. NaHCO_3_ and brine, dried, and evaporated to give a brown oil. Flash chromatography (EtOAc-hexanes-Et_3_N (40:55:5)) afforded **23** (283 mg, 91%) as a pale viscous oil; ^1^H NMR (500 MHz) *δ* 7.75 (1H, d, *J* = 8.9 Hz, H-6), 6.62 (1H, d, *J* = 8.9 Hz, H-5), 4.44 (2H, t, *J* = 8.1 Hz, H-2), 3.90 (3H, s, OMe), 3.79 (2H, s, NCH_2_CON), 3.46 (4H, s, 2 × NCH_2_Boc), 3.04 (t, *J* = 8.1 Hz, H-3), 2.81–2.93 (4H, m, NCH_2_CH_2_N), 1.45 (9H, s, CMe_3_), 1.42 (18H, s, CMe_3_). HRMS (FAB): Calcd for (C_31_H_49_N_4_O_10_ + H)^+^: 637.3449; found (M + H)^+^ 637.3425.

### 3.21. 2,2-((2-((Carboxymethyl)(2-(4-methoxy-7-nitroindolin-1-yl)-2-oxoethyl)amino)ethyl)azanediyl)diacetic acid (MNI-caged EDTA) ***24***

A solution of **23** (255 mg, 0.4 mmol) in TFA (20 mL) was stirred at rt for 4 h. The red-brown solution was concentrated in vacuo and the residue was dissolved in water (50 mL). The pH was raised from 1.22 to 7.36 via the careful addition of 1 M aq. NaOH and extracted with ether (3 × 50 mL). The aqueous solution (50 mL) was concentrated in vacuo, filtered through a 0.2 mm membrane and analysed via reverse-phase HPLC (mobile phase 15 mM Na phosphate, pH 6.0, 2 mM EDTA + 10% MeCN at 1.5 mL/min, at *t*_R_ 6.2 min). The aqueous solution was then diluted with water (800 mL, conductivity 125 mS) and purified via anion-exchange chromatography using a linear gradient formed from 10 to 250 mM TEAB (each 250 mL). Fractions containing the product, which were eluted at ~135 mM TEAB, were analysed as above. Pure fractions were combined, concentrated, and re-evaporated from MeOH. The residue was dissolved in water (20 mL) and the pH was raised from 5.58 to 7.36, filtered through a 0.2 mm membrane, and lyophilised. The residual powder was dissolved in water (7 mL) and quantified by means of UV spectroscopy: λ_max_ 330 nm (ε 4800 M^−1^cm^−1^), affording **24** (37.7 mM, 264 μmol, 66%). A fraction was exchanged to its sodium salt with Dowex-50. The sodium salt had ^1^H NMR (500 MHz D_2_O acetone ref.) 7.86 (1H, d, *J* = 9.0 Hz, H-6), 6.92 (1H, d, *J* = 9.0 Hz, H-5), 4.33 (2H, t, *J* = 7.9 Hz, H-2), 4.05 (2H, s, NCH_2_CON), 3.97 (3H, OMe), 3.88 (4H, s, NCH_2_), 3.67 (2H, s, NCH_2_), 3.51 (2H, t, *J* = 5.1 Hz, NC*H*_2_CH_2_N), 3.28 (2H, t, *J* = 5.1 Hz, NCH_2_C*H*_2_N), 3.12 (2H, t, *J* = 7.9 Hz, H-3). HRMS (FAB): Calcd for (C_19_H_24_N_4_O_10_ + Na)^+^, 491.1376; found: (M + Na)^+^ 491.1390.

### 3.22. Sequential Irradiations of NI-Caged BAPTA ***19***

Two separate solutions of 19 (0.5 mM)—one in 25 mM MOPS, pH 7.09 containing 5 mM dithiothreitol plus 2.5 mM EDTA and the other in 25 mM MOPS, pH 7.09 containing 5 mM dithiothreitol plus 2.5 mM Ca^2+^—were irradiated for increasing lengths of time (0, 0.5, 1, and 2 min) in 1 mm path-length cells in a Rayonet Photochemical Reactor (16 × 350 nm lamps). No photolysis was detected via UV spectroscopy in either solution (Figure 3). As control experiments, two separate solutions of methyl 1-[*S*-(4-amino-4-carboxybutanoyl)]-7-nitroindoline-5-acetate, *NI-caged glutamate* (0.5 mM) were irradiated under the same conditions described above. The expected 7-nitrosoindole (λ_max_ 413 nm) was observed, as previously reported [18] (data not shown).

### 3.23. Sequential Irradiations of MNI-Caged EDTA ***24***

Two separate solutions of **24** (0.5 mM) in 25 mM MOPS were prepared as follows; (A) plus 2.5 mM EDTA, pH 6.86; (B) plus 2.5 mM Ca^2+^, pH 6.86. Each solution was irradiated separately for increasing lengths of time (0, 0.5, 1, 2, and 4 min) in 1 mm path-length cells in a Rayonet Photochemical Reactor (16 × 350 nm lamps). UV spectra for each sample were recorded (Figure 4).

### 3.24. Estimated Extent of Photolysis of MNI-Caged EDTA ***24*** by HPLC

Two separate solutions of **24** (0.5 mM)—one in 25 mM MOPS, pH 7.02 plus 2.5 mM EDTA and the other in 25 mM MOPS, pH 7.09 plus 2.5 mM Ca^2+^—were irradiated for 0.25 min in 1 mm path-length cells in a Rayonet Photochemical Reactor (16 × 350 nm lamps). The extent of photolysis was determined via reverse-phase HPLC (mobile phase 15 mM Na phosphate, pH 6.0 + 10% MeCN, 1.5 mL/min). Quantification was based on peak heights compared to those of unphotolysed controls. The extent of photolysis of the solution containing 2.5 mM EDTA was 30.1% and that of the one containing 2.5 mM Ca^2+^ was 49.1%.

## 4. Conclusions

Our aim in exploring 7-nitroindolinyl-caged calcium chelators as tools for rapidly lowering Ca^2+^ concentrations in physiological media was to exploit the sub-microsecond time scale of photolysis of this caging technology. Previously reported examples [17,18] of caged calcium chelators exhibited photolysis rates that were significantly slower, at best in the ~250 µs range.

Synthesis of the nitroindolinyl-caged BAPTA **19** was relatively straightforward but lengthy, and ultimately fruitless, as it was not susceptible to photolysis either with or without the presence of calcium. In contrast, nitroindolinyl-caged EDTA **24** was readily accessed via a short synthetic route but clean photolysis to release free EDTA only occurred in the presence of saturating Ca^2+^, thus enabling no capacity to bind additional Ca^2+^. It appears that in both the BAPTA and EDTA cases, strong electron-donating centres are capable of quenching the excited state of the nitroindoline. Our experiments, and those of previous investigators, seem to have exhausted the options for conventional caging of calcium chelators. An intriguing possibility would be a photoisomerisable scaffold that could bring together two spatially separated iminoacetate groups, thereby assembling a complete hexadentate ligand. If this could be achieved, it would have an additional advantage of minimal Ca^2+^ affinity in the pre-irradiated form. Although attractive as a concept, present resources place this beyond our capability.

## Data Availability

The data presented in this study are available in Appendix A.

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
