# Peer review of "Synthesis and Photolytic Assessment of Nitroindolinyl-Caged Calcium Ion Chelators †"

_molecules, 2022, doi:10.3390/molecules27092645_

Round 1

Reviewer 1 Report

Comments to the Author

The authors synthesized low-affinity calcium chelators and caged them with Nitroindolinyl chromophore. The work is interesting to the readers in the field of organic photochemistry and sensor. Overall, the manuscript is interesting, and the experiments seem to be carried out carefully.  I recommended this article for publication.

But author need to include in the introduction “why low-affinity calcium chelators are important?” for Greater insight to the general readers.

The author needs to include the Kd value of Triacetic acid of EGTA and BAPTA in the manuscript.

Author Response

Thank you for your time in reviewing our manuscript and for your comments.

We have included a statement in the Introduction of the relevance of potential caged calcium chelators, and values for Kd of caged BAPTA and EDTA are now included in the manuscript.

Reviewer 2 Report

Dear Sir

The authors presented Synthesis and Photolytic Assessment of Nitroindolinyl-Caged Calcium Ion Chelators. The work is interested and can be accepted but the following comments should be considered before production

My comments

  • Aim of the work should be stated clearly in introduction
  • The purity of all chemicals should be mentioned with their originality
  • The current study should be compared with other related studies in literature
  • NMR charts should be supplied to supporting information

5- The authors must revise language of the manuscript before publication and the whole article must be adjusted based on journal style.

Author Response

Thank you for your time in reviewing our manuscript and for your comments.

We have included a statement of the relevance of this work in the Introduction. Purity of compounds is demonstrated by the inclusion of NMR spectra in the Supplementary Information Section and (for crystalline compounds) satisfactory elemental analysis. As is made clear in the Introduction, very little previous work on caged calcium chelators is available, and is referenced in the Introduction. As requested, we have amended the layout of the manuscript to follow journal style, but do not understand the request to revise the language of the manuscript, especially in the light of comments from Reviewers 1 and 3. We have published a number of previous papers on the chemistry of nitroindoline-caged compounds and have never previously been criticised for the writing style. Since no example of unacceptable style is given, it seems impossible to accede to the request.

Reviewer 3 Report

Photolytic behavior of organic components belongs to important research. Accurate description of results session in this manuscript could be a nice contribution to the field. I recommend publishing in current form.

Author Response

Thank you for your time in reviewing our manuscript and for your positive comments.
